# Stabilization of the Quadruplex-Forming G-Rich Sequences in the Rhinovirus Genome Inhibits Uncoating—Role of Na^+^ and K^+^

**DOI:** 10.3390/v15041003

**Published:** 2023-04-19

**Authors:** Antonio Real-Hohn, Martin Groznica, Georg Kontaxis, Rong Zhu, Otávio Augusto Chaves, Leonardo Vazquez, Peter Hinterdorfer, Heinrich Kowalski, Dieter Blaas

**Affiliations:** 1Center of Medical Biochemistry, Vienna Biocenter, Max Perutz Laboratories, Medical University of Vienna, Dr. Bohr Gasse 9/3, 1030 Vienna, Austria; martin.groznica@pasteur.fr (M.G.);; 2Institut Pasteur, CEDEX 15, 75724 Paris, France; 3Vienna Biocenter, Max Perutz Laboratories, Department of Structural and Computational Biology, University of Vienna, Campus Vienna BioCenter 5, 1030 Vienna, Austria; georg.kontaxis@univie.ac.at; 4Institute of Biophysics, Johannes Kepler University Linz, Gruberstr. 40, 4020 Linz, Austria; rong.zhu@jku.at (R.Z.);; 5Immunopharmacology Laboratory, Oswaldo Cruz Institute (IOC/Fiocruz), Av. Brasil, 4365, Rio de Janeiro 21040-360, Brazil

**Keywords:** picornavirus, rhinovirus, inhibition, RNA, G-quadruplex, pyridostatin, PhenDC3, folding, Na^+^, K^+^

## Abstract

Rhinoviruses (RVs) are the major cause of common cold, a respiratory disease that generally takes a mild course. However, occasionally, RV infection can lead to serious complications in patients debilitated by other ailments, e.g., asthma. Colds are a huge socioeconomic burden as neither vaccines nor other treatments are available. The many existing drug candidates either stabilize the capsid or inhibit the viral RNA polymerase, the viral proteinases, or the functions of other non-structural viral proteins; however, none has been approved by the FDA. Focusing on the genomic RNA as a possible target for antivirals, we asked whether stabilizing RNA secondary structures might inhibit the viral replication cycle. These secondary structures include G-quadruplexes (GQs), which are guanine-rich sequence stretches forming planar guanine tetrads via Hoogsteen base pairing with two or more of them stacking on top of each other; a number of small molecular drug candidates increase the energy required for their unfolding. The propensity of G-quadruplex formation can be predicted with bioinformatics tools and is expressed as a GQ score. Synthetic RNA oligonucleotides derived from the RV-A2 genome with sequences corresponding to the highest and lowest GQ scores indeed exhibited characteristics of GQs. In vivo, the GQ-stabilizing compounds, pyridostatin and PhenDC3, interfered with viral uncoating in Na^+^ but not in K^+^-containing phosphate buffers. The thermostability studies and ultrastructural imaging of protein-free viral RNA cores suggest that Na^+^ keeps the encapsulated genome more open, allowing PDS and PhenDC3 to diffuse into the quasi-crystalline RNA and promote the formation and/or stabilization of GQs; the resulting conformational changes impair RNA unraveling and release from the virion. Preliminary reports have been published.

## 1. Introduction

Rhinoviruses (RVs) cause common cold, an infection of the upper respiratory tract that usually takes a mild course [1,2]. However, RVs have been increasingly associated with more severe diseases such as bronchiolitis and pneumonia, and with exacerbation of chronic pulmonary disease, cystic fibrosis, and asthma [3]. The lost working days due to RV infections, along with the cost of symptom-alleviating medication, add up to billions of dollars per year in the USA alone [4,5]. Currently, there are no available RV vaccines or approved drugs against the 169 RV types https://www.picornaviridae.com/ensavirinae/enterovirus/enterovirus.htm (accessed on 18 March 2023).

RVs belong to the genus *Enterovirus* within the *Picornaviridae* family. Their icosahedral shell is built from 60 copies of each of the 4 capsid proteins (VP1 through VP4) and encloses a ~2.4 µm long ss (+) RNA genome of roughly 7200 nucleotides. The RNA is tightly folded to fit into the ~22 nm-diameter void in the protein shell; it carries a covalently linked 21-amino acid residue peptide (VPg) at the 5′ end and a poly-(A) tail at its 3′ end. Uptake into the cell occurs through receptor-mediated endocytosis. The receptor used depends on the RV species (A, B, or C) and the RV-receptor group (minor or major). Receptors include the low-density lipoprotein receptor (LDLR) and related proteins, intercellular adhesion molecule 1 (ICAM-1), and cadherin-related family member 3 (CDHR3) [6].

Once in the acidic environment of the host cell endosome, the native RV particle (~30 nm diameter) converts into the expanded (i.e., altered) A-particle (~31.2 nm diameter). Concomitantly, VP4 is lost, and the contacts between the RNA and the inner wall of the capsid change in preparation for genome exit [7,8]. On RNA egress, the empty B-particle remains. For the closely related poliovirus and enterovirus 71 [9,10], it was reported that structured regions of the viral RNA genome might transiently unfold during exit through a small hole in the A-particle; at least for RV-A2, RNA egress starts with the poly(A) tail [11,12]. Free VP4 might be forming a protective channel spanning the endosomal membrane for the safe transfer of the genome into the cytosol [13,14,15]. For other members of the genus *Enterovirus*, e.g., echovirus 18, evidence for genome release via reversible cracking of the capsid has been presented; here, exit is thought to occur through a much larger hole generated by the detachment of one or more of the 12 capsid pentamers [16]. A more recent cryo-EM analysis of RV-B14 showed that ~7% of the viral particles were found to be in this open conformation upon ICAM-1-triggered uncoating at low pH [17].

The development of the anti-RV compounds has focused on the capsid and the viral enzymes, as well as host factors essential for virus replication, e.g., Baggen et al. [18]. Capsid stabilizers (reviewed by Egorova et al. [19]), including the recently studied OBR-5-340 [20] and Compound 17 [21], prevent uncoating; however, all the potential antivirals suffer from the emergence of drug-resistant mutants. On the other hand, compounds acting on host factors may negatively affect vital cellular processes, leading to severe side effects; nevertheless, 2-deoxy-D-glucose showed promise by blocking the extensive energy generation that is required for viral synthesis [22].

Recently, nucleic acids have been identified as possible novel antiviral targets; molecules that bind structured domains in RNA genomes might impede conformational changes and thereby inhibit replication and translation [23]. Among these secondary structures are G-quadruplexes (GQs), which are 4-stranded higher-order folds formed intra- or inter-molecularly by guanine-rich stretches of DNA or RNA. The basic unit consists of a square, co-planar arrangement of four guanines (a tetrad or G-quartet) connected via Hoogsteen hydrogen bonds. GQs with at least two layers result from self-stacking via π-π interactions. Depending on the ionic radius, various cations can insert between these layers or in the center of the tetrads, leading to their stabilization. The core guanines are linked by three loops of varying sequence, topology, and size. The overall stability of individual GQs is governed by the number of layers, length, and identity of the loops, the flanking nucleotides, and the type of eventually bound cation. In contrast to DNA GQs, RNA GQs are practically monomorphic. RNA strands adopt a parallel orientation with a few exceptions [24,25], and they are preferentially stabilized by potassium ions binding between the stacked tetrads [26]. Higher-order structures may further originate from the 5′–5′ or 3′–3′ stacking of the terminal quartets of individual GQs.

GQs were identified in all domains of life and have been implicated in a large number of biological processes, including transcription, translation, epigenetic regulation, DNA recombination, splicing, and mRNA transport [27,28]. GQs are also found in the genomes of DNA and RNA viruses, such as the Ebola virus, herpes simplex virus, human papillomavirus, human immunodeficiency virus, Zika virus, influenza virus, human coronaviruses, and hepatitis C virus [29,30,31,32,33,34,35,36], where they control various steps in the virus life cycle, ranging from protein expression and nucleic acid replication to assembly into nucleocapsids [37]. An increasing number of GQ-stabilizing compounds, such as pyridostatin (PDS), PhenDC3, Braco19, and many others, have been synthesized. They inhibit the replication of various viruses at the level of transcription, translation, and/or genome replication via stabilizing GQs and acting as physical roadblocks for viral DNA and RNA polymerases or host ribosomes. Such compounds may also compete with host-cell or virus proteins for binding to the above GQs [38].

A previous bioinformatics analysis indicated that putative GQ structures are present in the RNA genomes of picornaviruses, including all RVs [39] and enterovirus A71 [40]. However, their possible functional role remains untested. Using the QGRS Mapper, a G-quadruplex prediction tool, we confirm that GQs are potentially present in the genomes of all three rhinovirus species (A, B, and C). The predicted structures might mostly consist of two tetrads, and some of the GQs might contain one zero-nucleotide loop. However, as stressed by various authors, prediction alone proves neither the in vitro nor the in vivo presence of GQs [41,42], and confirmation by orthogonal assays is needed. Using RV-A2 as a model, we demonstrate that two putative Quadruplex-forming-G-Rich-Sequences (QGRSs) might fold into two-layer GQ structures under physiological conditions. In infectivity studies, we find that the GQ-binding compounds, PDS and PhenDC3, inhibit viral RNA uncoating likely by promoting the conversion of metastable alternative structures in the viral RNA into GQs. Evidently, this entirely depends on the access of the compounds to the target sequences in the genomic RNA, which is determined by the degree of compaction of its tertiary structure. The latter appears to be governed by the respective Na^+^ and K^+^ concentrations. In vivo, the extracellular and intracellular phases of the virus replication cycle expose the virions to changes in these ionic environments.

Previous analyses of other viruses (see above) indicated that GQ-stabilizing compounds interfere with the viral life cycle following the release of the genetic material from the nucleocapsids. Conversely, our results with RV-A2 rather suggest that, at least in RVs, G-rich sequences with the potential to form GQs might be targeted while still confined in the viral capsid, highlighting the impact of the different ions on viral uncoating.

## 2. Materials and Methods

### 2.1. Bioinformatics Analysis of Rhinoviral Genomes for Identification of Putative GQs

The QGRS mapping software was kindly made available by Paramjeet S. Bagga from Ramapo College, New Jersey. The underlying algorithm [43] predicts the likelihood of a sequence containing G-repeats to form a stable GQ, the stability score (G-score), based on published biophysical data. We performed our search using the default search options: maximal length = 45, minimum G-group = 2, and loop size ≥0 and ≤36. Allowing a loop size of 0 (zero loop) includes unconventional QGRS, as reported in yeast microsatellite DNA and mRNA involved in polyamine biosynthesis [44,45]. Complete RV-A (75), RV-B (26) and RV-C (19) sequences (120 sequences in total available at the time of writing and accessible through The Pirbright Institute) were used. Non-overlapping sequences with a G-score ≥10 within a sliding window of 20 residues were rendered as a 2D-line plot representing the QGRS length; the color bar refers to the individual G-scores. We wish to point out that the G-scores provided by the downloadable C++ implementation of the QGRS mapping algorithm used here occasionally differ from those obtained with the web-version https://bioinformatics.ramapo.edu/QGRS/analyze.php (accessed on 18 March 2023) of the QGRS Mapper (see note on Github, https://github.com/freezer333/qgrs-cpp; accessed on 18 March 2023).

### 2.2. Oligonucleotides and QG-Binding Compounds

A synthetic 27-mer RNA oligonucleotide representing human telomeric repeat-containing RNA (miniTERRA [46]) was used as a positive control; the RV-A2 sequence-derived G20 and G11 and the negative control miniTERRA with all Gs replaced by Cs, are listed in Table 1. Real-time PCR primers (RV-A2 Fw: 5′ gccccatgtgtgcagagttttc 3′; Rv: 5′ aggtgtcagtgttatttattggtactaggctg 3′ and Aichi virus A (AiV) Fw: 5′ tgtacaacacccactccatgtg 3′; Rv: 5′ tccacagagagggagttcctg 3′) were purchased from Microsynth. Thioflavin T (ThT; Merck) was prepared as a 250-µM stock solution in water and kept at 4 °C. Pyridostatin (PDS; Merck) was prepared as a 20 mM stock solution in water. PhenDC3 (Merck) was prepared as 1 mM stock solution in DMSO. The latter two compounds were aliquoted and kept at –80 °C, thawed, and diluted immediately prior to use.

### 2.3. Cells and Virus

HeLa Ohio cells, for simplicity called ‘HeLa’ throughout this article, were originally obtained from ATCC and maintained in DMEM (Merck), supplemented with 10% FBS (Life Technologies, Carlsbad, CA, USA), and 1% penicillin (Merck, New York, NY, USA) and streptomycin (Merck). Cells were kept in a humidified, 5% CO_2_-containing atmosphere at 37 °C. For infection, the serum concentration was reduced to 2% FBS, and cells were incubated at 34 °C, the optimal growth temperature for RVs. RV-A2 was initially acquired from ATCC, propagated, and purified following the protocol detailed in [47].

### 2.4. ThT Assay for Detection of GQ

Ribooligonucleotides were dissolved in 100 mM K^+^ or Na^+^ phosphate buffer (pH 7.4) to a final concentration of 5 µM, incubated for 10 min at 90 °C, followed by 10 min at 4 °C, and mixed with ThT to a final concentration of 5 µM. The samples were excited at 440 nm, and emission was measured at 490 nm using a PerkinElmer VICTOR Nivo Multimode Plate Reader (Waltham, MA, USA). The signal was acquired at 34 °C.

For the preparation of viral RNA cores (‘ex-virion RNA’), the protein shell of purified RV-A2 (~2 µg) was digested with 5 µg proteinase K overnight at 4 °C in a final volume of 10 µL. The ex-virion RNA samples were ultrafiltered using 100 K Merck Amicon Ultra Filter units according to the manufacturer’s protocol, followed by four 400 µL washes with 100 mM Na^+^ or K^+^ phosphate buffer (pH 7.4). The samples were mixed with ThT (final concentration of 5 µM), and the volume was adjusted to 100 µL with the respective buffers. The ThT fluorescence signal was acquired as described above, at 30 °C. The same samples were then incubated for 10 min at 60 °C followed by cooling to 30 °C over 30 min, and ThT fluorescence was again measured.

### 2.5. Fluorescent Indicator Displacement Assay (FiD)

Ribooligonucleotide samples were prepared as above and supplemented with ThT to a final concentration of 5 µM. Then, 100 µL was dispensed into the wells of a 96-well plate. PDS was then added every 30 s in 2 µM steps up to 50 µM and the respective fluorescence was recorded at 34 °C. Signal loss due to ThT dilution by the solvent (water) was negligible. A titration curve was constructed from three independent experiments by plotting the percentage fluorescence drop obtained by dividing the mean fluorescence intensity at each PDS concentration by the mean initial signal (no PDS x 100) against the concentration of PDS. Individual IC_50_ values were calculated for each condition by non-linear fitting using GraphPad Prism 8.4.3 (San Diego, CA, USA).

### 2.6. The Particle Stability Thermal Release Assay (PaSTRy)

Ref. [48] was performed according to Real-Hohn et al. [49] with minor adaptations. RNA accessibility was monitored with SYTO 82 (Thermo Fisher, Waltham, MA, USA) in a Bio-Rad CFX Connect Real-Time PCR instrument (Hercules, CA, USA). Purified RV-A2 (~3.5 µg) in PBS minus PDS (control) and plus PDS at 200 µM final concentration, was incubated for 4 h at 4 °C (negligible virus breathing) or 34 °C (strong virus breathing). Unbound PDS was removed by ultrafiltration in 100 K Merck Amicon Ultra Filter units, followed by four PBS washes to eliminate the remaining unbound PDS. SYTO 82 was added to a final concentration of 5 µM, and the volumes were adjusted to 70 µL with PBS. Three 20 µL aliquots from each of these samples were dispensed into the wells of a thin-walled PCR plate, the temperature was ramped from 25–95 °C at 1.5 °C/min, and SYTO 82 light-up fluorescence was recorded. Six independent measurements were made for each condition. Data were rendered as a dot plot revealing the temperature at which the RNA becomes accessible for SYTO 82 binding [49].

### 2.7. Melting Analysis of RV-A2 ‘Ex-Virion’ RNA

RNA cores in 100 mM Na^+^ or K^+^ phosphate buffer at pH 7.4 were prepared as in Section 2.4. SYTO 82 was added to a final concentration of 5 µM, and the volumes were adjusted to 70 µL with the respective buffers. Three 20 µL aliquots from each of these samples were dispensed into the wells of a thin-walled PCR plate, and the temperature was ramped from 25–95 °C by 1.5 °C/min, and SYTO 82 light-up fluorescence was recorded. Three independent measurements were performed for each condition; means of fluorescence signals are depicted.

### 2.8. Virus Yield Assay

Purified RV-A2 (~1 µg) +/− PDS at a 20 µM final concentration in 100 µL of 100 mM Na^+^ or K^+^ phosphate buffer (pH 7.4) was incubated overnight at 25 °C. Unbound PDS was removed by ultrafiltration as described above. Following infection of HeLa cells with the retentate, virus titers (TCID_50_) were determined as described elsewhere [50].

### 2.9. Immunocytochemistry and Flow Cytometry

Cells grown in a 6-well plate until 90% confluent were infected at MOI = 1 with RV-A2 that had been preincubated for 4 h at 34 °C in plain PBS (control) or in PBS containing PDS at 20 or 200 µM and subjected to ultrafiltration. An identical experiment was carried out by using PhenDC3 at 1 µM and 5 µM instead of PDS. The cells were infected with the retentate of the ultrafiltration and 9 h post-infection (pi) they were washed once with PBS and detached with 0.1% trypsin and 0.05% EDTA in PBS (Merck). Trypsin was inactivated (in 10% FBS in DMEM), and cells were harvested by centrifugation at 300× *g* for 3 min at 4 °C. The pellet was resuspended in 500 µL ice-cold PBS, an equal volume of 4% formaldehyde in PBS was added, and the mixture was incubated for 10 min at 4 °C. This and all subsequent steps were done with gentle rocking at 4 °C. Cells were washed three times with PBS plus 0.1% Tween-20 (PBST), resuspended in PBS + 0.1% Triton X-100, and incubated for 10 min. The fixed cells were then incubated in blocking buffer (PBS, 1% BSA, 0.1% Tween-20) for 30 min. Then, 10 µg/mL of 8F5, a monoclonal antibody (mAb) specific for VP2 of RV-A2 [51], was added, and incubation continued for 1 h. Cells were again washed three times with PBST and incubated for 1 h with goat anti-mouse AlexaFluor 488 antibody (Thermo Fisher) diluted (1:1000) in blocking buffer, then incubated with Hoechst 33342 solution (Thermo Fisher) in PBS (1:2000) for 10 min for staining nuclei, followed by three PBST washes. Cells were resuspended in PBS and analyzed with a BD Bioscience FACSAria III flow cytometer; more than 10^4^ events were acquired for each sample. Forward scattering (FSC) v/s VP2 (FITC-A) plots were generated by Tree Star FlowJo X v10.0.7 software (Ashland, OR, USA).

Cells were cultured until ~80% confluent and infected with RV-A2 pretreated +/− 200 µM PDS (as for flow cytometry, see above); at 30 min pi, the medium was replaced with PBS. Cells were gently detached and subjected to three freeze–thaw cycles. Cell debris was removed by low-speed centrifugation, and half the sample was used to analyze viral uncoating intermediates, i.e., subviral A-particles plus (empty) B-particles. Particles were immunoprecipitated using mAb 2G2 [52] bound to protein G magnetic beads (Dynabeads-Protein G; Life Technologies). The rest of the sample was processed identically, but mAb 2G2 was omitted as a negative control. As a positive control, ~1 µg of heated RV-A2 (10 min at 56 °C), resulting in almost ~100% conversion into subviral B-particles, was processed identically. After extensive washing with PBS, the immunoprecipitates were resuspended in 100 μL PBS. Then, 2 µL of 5× protein sample buffer was added to 18 µL of each sample, and the mixture was heated to 95 °C for 10 min. Proteins were separated by SDS-PAGE (10%) followed by electro-transfer to an Immobilon-P membrane (Millipore, Burlington, MA, USA), essentially as described in [53]. In brief, VP2 was detected with mAb 8F5, goat anti-mouse-horseradish peroxidase (Thermo Fisher), and SuperSignal West Pico PLUS chemiluminescent substrate (Thermo Fisher). The signal was quantified using a ChemiDoc Gel Imaging System (Bio-Rad). To determine the proportion of A- and B-particles, the viral RNA was also quantified. As an internal control and for normalization, 100 µL Aichi virus (AiV; a member of the kobuvirus species of the *Picornaviridae* family), corresponding to 2 × 10^7^ TCID_50_, was added to each sample. RNA was recovered using TRIzol (1 mL; Invitrogen, Waltham, MA, USA) with extraction and precipitation (together with GlycoBlue from Invitrogen) following the manufacturer’s protocol. First-strand cDNA synthesis was carried out with the NEBNext reagent kit (New England Biolabs, Ipswich, MA, USA) using random primers and quantified by qPCR using primers specific for RV-A2 or for AiV in a CFX96 Touch Real-Time PCR Detection System (Bio-Rad).

### 2.10. Sucrose Density Gradient Sedimentation

Cells were infected with RV-A2 pretreated +/− 200 µM PDS as described above, the medium was replaced with PBS, and the resuspended cells were subjected to three freeze–thaw cycles. Cell debris was removed by low-speed centrifugation. Then, 500 µL of the resulting supernatants containing viral material were deposited onto preformed 10–40% (*w*/*v*) sucrose density gradients made in virus buffer (50 mM NaCl, 20 mM Tris-HCl, pH 7.4) and centrifuged at 4 °C for 30 min in an SW55 Ti rotor (Beckman, Pasadena, CA, USA) at 287,000× *g*. Aliquots (250 μL) were collected from top to bottom, and 20 μL of each fraction was deposited onto a methanol-activated Immobilon-P membrane in a 96-well Bio-Dot Microfiltration Apparatus (Bio-Rad). VP2 was detected using VP2-specific mAb 8F5 [51] and IRDye 680RD goat anti-mouse IgG secondary antibody (LI-COR), essentially as described for Western blotting above. The fluorescent signal was acquired with an Odyssey Infrared Imager (LI-COR).

### 2.11. Time-of-Drug-Addition Experiments

HeLa cells grown in 6-well tissue culture plates to roughly 80% confluency were challenged with RV-A2 at MOI = 10 for 30 min in infection medium at 4 °C with steady rocking, allowing the virus to attach to its receptor while preventing its internalization, achieving synchronized infection. The inoculum was removed, cells were washed three times with PBS, a fresh infection medium was added, and incubation continued at 34 °C to allow for virus internalization. Immediately (=0 min) and 180 or 300 min pi, PDS was added to a final concentration of 20 µM. At 9 h pi, cells were processed for flow cytometry and immunocytochemistry as above. Using Tree Star FlowJo X v10.0.7 software, a FITC-A histogram showing the mean fluorescence intensity (MFI) from more than 10^4^ events corresponding to de novo produced VP2 was generated upon gating based on FSC and SSC properties.

### 2.12. Electron Microscopy and Rotary Shadowing

Ex-virion RNA in 100 mM Na^+^ or K^+^ phosphate buffer (pH 7.4) was prepared as detailed in Section 2.4. The RNA samples were first diluted to ~0.1 mg/mL in the absence (control) or presence of 20 µM PDS and incubated for 10 min at ambient temperature. The samples were subsequently diluted 1:1 in spraying buffer (200 mM ammonium acetate and 60% (*v*/*v*) glycerol, pH adjusted to 7.4). Samples were immediately sprayed onto freshly cleaved mica chips (Agar Scientific, Birchanger, Essex, UK) and quickly transferred into a BAL-TEC MED020 high vacuum evaporator equipped with electron guns. While rotating, samples were coated with 0.6 nm platinum (BALTIC) at an angle of 7°, followed by 6 nm carbon (Balzers, Liechtenstein) at 90°. The replicas were floated off the mica chips, picked up on 400-mesh Cu/Pd grids (Agar Scientific), and inspected in an FEI Morgagni 268D TEM (Thermo Fisher Scientific) operated at 80 kV. Images were acquired using an 11-megapixel Morada CCD camera (Olympus SIS, Münster, Germany).

### 2.13. Atomic Force Microscopy (AFM)

Ex-virion RNA was prepared as above. The samples were incubated with 20 µM PDS for 10 min, then deposited onto freshly cleaved mica and immediately imaged in a Pico-SPM atomic force microscope (Molecular Imaging) equipped with a fluidic cell. The AFM images were acquired with acoustic AC (tapping mode) using the MSNL (Bruker, Billerica, MA, USA) cantilever E (with a nominal spring constant of 0.1 N/m) at 15 kHz. The scanning speed was 3000 nm/s, and the number of pixels per line was 256.

### 2.14. Quantification of PDS Binding to the Virus

Purified RV-A2 (12 µg) was incubated with 20 µM PDS in 100 mM K^+^ or Na^+^ phosphate buffer (pH 7.4) for 4 h at 4 °C (negative control with no virus breathing) or 34 °C (permitting virus breathing) [54,55]. Unbound PDS was removed by ultrafiltration in 100 K Merck Amicon Ultra Filter units, followed by 4 × 400 µL washes with the respective buffers. Virus samples were then incubated with methanol containing 1% formic acid. Then, 1 µL of the extract was injected into an UltiMate 3000 RSLC system (Thermo Fisher) to denature the capsid and allow PDS extraction from within the virion. After centrifugation, the supernatant was analyzed with liquid chromatography-tandem mass spectrometry (Thermo Fisher Scientific) directly coupled to a TSQ Vantage mass spectrometer (Thermo Fisher Scientific) via electrospray ionization in the positive ion mode. A Kinetex C18 column (100 Å, 150 × 2.1 mm) was used beforehand to separate PDS from other components of the extract, employing a flow rate of 80 µL/min. A 10-min linear gradient was used from 95% A (1% acetonitrile, 0.1% formic acid in water) to 80% B (0.1% formic acid in acetonitrile). LC-MS/MS was performed by employing the selected reaction monitoring (SRM) mode of the instrument and using the transitions 597.4 *m*/*z* → 511.3 *m*/*z* (CE 20) and 597.4 *m*/*z* → 468.3 *m*/*z* (CE 25). Data were interpreted manually, and the absolute amount of PDS was quantified by using a calibration curve obtained with pyridostatin standard solutions.

### 2.15. Quantification and Statistical Analysis

At least two experimental and biological replicates were performed. The data are displayed as mean ± standard deviation (SD) for *n* > 2. Statistical significance was determined using the unpaired, one-tailed Student’s *t*-test. The *p*-value and sample size *n* are provided in the respective figure legends.

## 3. Results

### 3.1. QGRS Mapper Identifies Potential GQ-Forming Sequences in All RV Genomes

Using the C++ implementation (https://github.com/freezer333/qgrs-cpp; accessed on 18 March 2023) of the QGRS Mapper algorithm [43], we investigated all RV complete genome sequences available in GenBank. As RNA GQs are generally more stable than the corresponding DNA GQs [56], we adopted the search motif G ≥ 2L0-36G ≥ 2L0-36G ≥ 2L0-36G ≥ 2L0-36 (maximum length 45 bp), allowing long and zero loop GQs and two-layer GQs, as both features are common [44], and plotted the G-score (threshold ≥10) against the position of the first nucleotide of the respective putative QGRS in the genomic RNA sequences (Figure 1).

Assuming that RNA almost exclusively adopts parallel GQs, G11 would give rise to an unusual monomeric two-layer GQ bearing a zero-nucleotide loop 3 in combination with a long loop 1 (Appendix A, top).

### 3.2. Thioflavin T Light-Up Assay and Competition with PDS Demonstrate Specific Binding

We then used a thioflavin T (ThT) light-up assay to assess whether the two RV-A2-derived RNA sequences (G11 and G20) and the positive control miniTERRA, a sequence derived from Telomeric Repeat-containing RNA, form GQ structures at room temperature. ThT end stacks with RNA GQs resulting in highly increased fluorescence intensity [57]. As seen in Figure 2A,B (left panels), no significant fluorescence was detected upon incubation of ThT with the negative control (see Table 1). In contrast, a clear signal was recorded for all three tested QGRS, regardless of whether the measurement was in Na^+^-or K^+^-containing buffer. It was slightly lower for G11 in Na^+^ phosphate buffer (Figure 2A) when compared to K^+^ phosphate buffer (Figure 2B). For miniTERRA and G20, it was between 1.5 and 2-fold higher in the presence of K^+^ phosphate buffer (left panel of Figure 2B). A similar ion dependency was previously observed for 18S rRNA-derived 2- and 3-layer GQ-forming sequences [58]. The ThT fluorescence intensity was strongest for miniTERRA and approximately 3–4-fold lower for G11 and G20 (but still 30–40 times higher than for the negative control, Figure 2A,B; left panels).

PDS binds more strongly to GQs than ThT and displaces the latter from its target. We thus examined whether PDS also competes with ThT for the above sequences and whether Na^+^ or K^+^ impacts PDS interaction with the respective GQs by using a Fluorescent indicator Displacement (FiD) assay similar to the one described by Mestre-Fos and colleagues [58]. The dose-response curves for miniTERRA and G20 were similar in Na^+^ and K^+^ phosphate buffers (Figure 2A,B, right panels). In contrast, for G11, the inhibitory concentration reducing binding to 50% (IC_50_) was almost double in the presence of Na^+^ when compared to K^+^ (Table 2); notably, for the lower G-scoring G11, a sharp decrease in the ThT fluorescence signal was already evident at low PDS concentrations, indicating lower ThT binding affinity.

### 3.3. PDS and PhenDC3 Reduce RV-A2 Infectivity

The prevailing model of RV uncoating proposes a requirement for reorganization and transient unfolding of the genomic RNA [7,9,11,12,59]. We reasoned that diffusing PDS into the virions to allow its binding to the RNA might stabilize existing GQs and convert G-rich sequences into GQs, thereby interfering with RNA release. To enable access of PDS and PhenDC3 to the viral RNA, we exploited temperature-dependent capsid breathing, a dynamic expansion of the native virion that occurs between 25 °C and 37 °C [60].

Purified RV-A2 was incubated for 4 h with PDS at 4 °C, where capsid breathing is strongly reduced, and at 34 °C, which is also the optimal growth temperature of RVs [61,62]. Unbound PDS was removed, and PaSTRy [48,49] was performed to determine whether PDS impacts heat-induced in vitro uncoating [63]. Plotting degrees C versus fluorescent emission reveals the temperature where genomic RNA becomes accessible to SYTO 82 as the first inflection point in fluorescence intensity [49], Appendix A. When RV-A2 was pre-incubated with PDS at 34 °C, SYTO 82 gained access to the RNA already at a lower temperature (T_on_ = 37.7 °C) when compared to control conditions (T_on_ = 42.4 °C), i.e., pre-incubation with PDS at 4 °C, which prevents the dynamic opening of pores in the capsid, and pre-incubation at 4° C, as well as at 34 °C, without PDS (Figure 3A). However, the peak of the SYTO 82 signal indicates the complete conversion of native virions into the porous A-particles (T_max_), and the peak of the first derivative (at T_50_, i.e., 50% RNA accessibility) [49] remained essentially unaltered (Appendix A, lower panel).

We then assessed whether PDS influenced viral attachment and/or in vivo uncoating. RV-A2 was pre-incubated with PDS and without PDS (control) at 34 °C for 4 h, as described above. Unbound PDS was removed by ultrafiltration in an Amicon ultrafilter unit, followed by repeated washing with PBS at 25 °C. HeLa cells were then incubated with these virus samples for 30 min at 34 °C to allow for viral uptake and uncoating. The cells were broken, debris removed, and the immunoprecipitates were collected with mAb 2G2, which exclusively recognizes A-particles ([VP1VP2VP3]60 + ssRNA) and (empty) B-particles ([VP1VP2VP3]60) [52,64]. Protein and RNA from the precipitated material were separately quantified using densitometry of Western blots (Figure 3B) and RT-qPCR (Figure 3C). Regardless of whether the pre-incubation was carried out in the presence or absence of PDS, the same quantity of viral protein (using VP2 as readout) was detected. This excludes the possibility that PDS impacts cell attachment or the overall rate of conversion of the native virus into subviral (A + B) particles. Aggregation of PDS into long filaments, as exclusively observed in the Tris-HCl buffer, can also be excluded [65] because Tris was never present in these assays. However, there was 70% more viral RNA in the sample pre-incubated with PDS, indicating that significantly less RNA was released from the virions during the first 30 min pi (Figure 3C).

To confirm this result by using an orthogonal assay, HeLa cells were infected with native and PDS-treated RV-A2, and 30 min pi cell-associated viral material was released by freeze-thawing, as described above. The clarified cell extract was analyzed by sedimentation through preformed sucrose density gradients. Sucrose gradient ultracentrifugation separates native virions, A-particles, and empty B-particles [66]. The virus incubated without PDS (control) resulted in a peak at 80S, indicating near-complete conversion into B-particles (Figure 3D). However, the virus incubated with PDS resulted in a substantial fraction sedimenting between 150S (native virus) and 80S (empty particles), corresponding to RNA-containing A-particles (hash). These results confirm that RNA release from the virus is impaired in vivo when PDS is present in the capsid.

Reduced RNA release is likely due to the PDS-mediated formation and/or stabilization of GQs in the encapsidated viral RNA. We further validated this assumption by using PhenDC3, another frequently employed GQ-binding compound [67]. The HeLa cells were challenged with RV-A2 pre-incubated as above with PDS (20 µM and 200 µM) and in parallel with RV-A2 pre-incubated with PhenDC3 (1 µM and 5 µM). The mock-treated virus was used as a control. After one infection cycle (9 h pi), RV-A2-positive cells were determined by fluorescence-activated cell sorting (FACS) using intracellular VP2 as a readout for replicated RV-A2. Upon pre-incubation with PDS (20 µM and 200 µM) as well as with PhenDC3 (5 µM and 1 µM), we observed a concentration-dependent decrease in the fraction of cells containing *de-novo* synthesized RV-A2 (Figure 3E; note that an effect of PhenDC3 was only apparent at 5 µM). These data further strengthen our hypothesis that GQ stabilization triggers structural changes in the encapsidated RNA that reduce genome release in vivo and consequently virus production.

### 3.4. PDS Affects the Conformation of RV-A2 Genomic RNA and Reduces Viral Infectivity in the Presence of Na^+^ but Not K^+^

We then studied whether PDS induced any structural changes in protein-free RV-A2 cores under different ionic conditions. Encapsidated RV RNA exhibits a quasi-crystalline packing [68] with its phosphates mainly neutralized by K^+^; based upon data on the related poliovirus, approximately 7500 negative charges are neutralized by about 4900 K^+^, 900 Na^+^, 120 Mg^2+^, polyamines, and basic amino acids per virion [69,70]. However, once the virus is released from the host cell, this proportion of ions should shift in favor of Na^+^, which prevails in the extracellular medium, since the capsid is permeable to such ions [71].

Adding ThT to protein-free virion cores at 30 °C resulted in increased fluorescence emission. The signal was ~70% higher in Na^+^ phosphate as compared to K^+^ phosphate buffer (Figure 4A, 30 °C, upper panel). This contrasts with the stronger ThT fluorescence observed for GQ-forming ribooligonucleotides in K^+^ (Figure 2 and [72]) and rather suggests that the protein-free viral core may be more compact in the presence of K^+^, leading to reduced accessibility of pre-existing GQs and QGRS by the ThT probe. Even more remarkable, the ThT emission intensities recorded for identical samples heated to 60 °C followed by slow cooling to 30 °C were about 20 times higher (Figure 4A, 60 °C > 30 °C; lower panel; note the different scales). This may be taken to indicate that most QGRS within the RV-A2 genome were trapped in kinetically competing structures. Heating of the ex-virion RNA might resolve these metastable alternative structures (likely hairpins), transforming them into the more stable GQs during cooling, as evidenced by increased ThT emission. Note that ThT-binding is now much less affected by the monovalent cation type (just by 9%; note the different scales). The markedly decreased difference between the effects of Na^+^ and K^+^ might be due to ThT being present in the sample during the viral RNA transitions from a less structured state at 60 °C to a tighter conformation at 30 °C.

To assess the ‘degree of compactness’, we incubated RNA cores in Na^+^ or K^+^ phosphate buffer +/−PDS and submitted them to rotary shadowing. The platinum replicas were examined by transmission electron microscopy (TEM) (Figure 4B). In the absence of PDS, the cores appeared approximately spherical in K^+^ and somewhat more flattened and prolate in Na^+^ (compare the insets), suggesting that Mg^2+^ and polyamines that are bound to the encapsidated genome [69,70,73] were still present, maintaining the RNA in the compact form it adopts inside the virion. The addition of PDS to 20 µM in the Na^+^ phosphate buffer led to the elongation and expansion into irregular rods (Figure 4B, left panels). These data are consistent with our PaSTRy results, which also support PDS-induced structural reorganization (Figure 3A). No such effect of PDS was seen in the K^+^ phosphate buffer, where the ex-virion RNA remained roughly spherical (Figure 4B, right panel).

To support these TEM results, we used atomic force microscopy (AFM). AFM scans of samples prepared as above yielded very similar images (compare Figure 4B,C). Again, in the presence of PDS, we observed mostly irregular rods with lengths of up to 200 nm and a diameter of about 10 nm in Na^+^ phosphate buffer, while more spherical objects with a diameter of ~25 nm were seen in K^+^ phosphate buffer (Figure 4C); this corresponds roughly to the dimension of the capsid cavity. Lower magnification overviews are shown in Appendix A. Taken together, the AFM analysis confirmed that the structures shown in Figure 4B represent differently compacted viral RNA and are not artifacts of the platinum contrast and/or the drying process.

The drastic shape change in the RNA induced by PDS and only observed in Na^+^ phosphate buffer may be triggered by the refolding of metastable hairpin-like structures, involving part or all predicted QGRS, into PDS-stabilized GQs, determining the global structure of the RNA genome within the protein shell [74,75]. The increased compaction caused by K^+^ might limit the access of PDS to these metastable regions, accounting for the different effects of Na^+^ and K^+^.

To substantiate the above hypothesis, we carried out a melting analysis of the ex-virion RNA, similarly as described by Silvers et al. [76], exploiting SYTO 82 for monitoring structural transitions of the RNA via fluorescence intensity. SYTO 82 shows a bright emission on intercalation into nucleic acid duplexes with no sequence preference, and it does not influence their thermal stability at the used concentration [77]. In native RNA, the binding of intercalating dyes to helical segments is restricted by the tertiary structure, e.g., refs [78,79]. Fluorescence traces obtained in Na^+^ and K^+^ phosphate buffers suggest that access of SYTO 82 to the RNA cores was restricted between 25 °C and 40 °C. This implies that the genome remains compact within this temperature range (Appendix A). The subsequent rapid increase in the fluorescence from the low-temperature baseline (at 44.5 °C and 51 °C for Na^+^ and K^+^, respectively) to the fluorescence peak (=maximal response; at 54.3 °C and 57.1 °C for Na^+^ and K^+^, respectively) likely results from disruption of the RNA tertiary structure, allowing SYTO 82 to bind newly exposed helices originally buried in the RNA core. The rise was ~2-fold (Na^+^) and ~1.2-fold (K^+^), suggesting a more extensive loss of tertiary organization in the Na^+^-phosphate buffer. The higher tertiary structure stability in K^+^ phosphate buffer is illustrated by the approximately 4 °C higher melting temperature T_m1_. The following progressive drop in emission intensity is due to the unraveling of more stable secondary structures releasing the SYTO 82; the corresponding T_m2_ for this transition was 60.5 °C in Na^+^ and 63 °C in the K^+^-containing buffer.

Altogether, the above in vitro data on protein-free viral RNA cores substantiate the idea that K^+^ imposes a tighter tertiary structure on the viral genome, thereby limiting access of PDS to putative internal QGRS and preventing GQ formation/stabilization. If this is correct, a reduced number of PDS molecules should associate with the RV-A2 genome in a K^+^ phosphate buffer when compared to a Na^+^ phosphate buffer. Since assessing this figure is technically less demanding with encapsidated RNA, we quantified PDS levels in virions after incubation with PDS at 20 μM at 4 °C or at 34 °C for 4 h in Na^+^ or K^+^ phosphate buffers. The virus was extensively washed by ultrafiltration to remove all traces of free PDS. PDS remaining associated with the virus was then quantitated by mass spectrometry. In agreement with the above results, the number of PDS molecules was highest upon incubation in Na^+^ phosphate buffer at 34 °C (Table 3).

This result corroborates our hypothesis that K^+^, while usually cooperating with PDS in GQ stabilization [80], prevented PDS from diffusing into the tight RNA core, regardless of whether it was protein-free or encased within the viral shell rendered permeable at 34 °C. Thus, the differential effect of Na^+^ versus K^+^ on PDS-binding to viral QGRS was by far overcompensated by their differential effect on the permeability of the RNA core, which is indispensable for the PDS to attain its targets. In agreement with the above, pre-incubation of RV-A2 with PDS in K^+^ phosphate buffer at 34 °C followed by the challenge of cells had no significant influence on the final viral yield (Figure 4D; K^+^), whereas the same pre-incubation in Na^+^ phosphate buffer resulted in a roughly 100-fold decrease (Figure 4D, Na^+^).

Finally, we wanted to identify the step of the RV-A2 infection cycle that was affected by PDS in vivo (i.e., without pre-incubation) by a time-of-addition experiment similar to that described for Zika virus infection of Vero cells [81]. The virus was bound to the cells for 30 min at 4 °C, PDS was added to a 20 µM final concentration, and the cells were shifted to 34 °C (=0 min pi). In parallel, PDS was added at 180 min and at 300 min pi. These times correspond roughly to entry and uncoating (0–180 min), the peak of RNA synthesis (180–300 min), and virus assembly (from 300 min onwards) [82,83,84]. The cells were maintained for 9 h pi (one infection cycle), and de novo viral synthesis was measured by fluorescence-activated cell sorting (FACS). As seen in Figure 4E, the percentage of infected cells was impacted only when PDS was added at T = 0 min pi; this reinforces the role of Na^+^ in PDS-mediated viral inhibition, as the Na^+^ concentration is highest in early endosomes (reviewed in Scott and Gruenberg [85]). This is in stark contrast to the Zika virus, where a comparable concentration of PDS specifically affected post-entry viral RNA replication and protein expression [81].

## 4. Discussion

QGRS Mapper predicted conserved G-rich sequences that potentially fold into GQs in all RV genomes (Figure 1); in the majority of the cases, they would give rise to two-layer G-quartets. Four of these putative QGRS are highly conserved in species A, B, and C and were also identified in the bioinformatics analysis by Lavazzo et al. using an in-house algorithm [39]. While this conservation suggests functional relevance, it is possible that it only reflects an important function in the encoded protein.

We chose RV-A2, one of the best-characterized RVs [86], for experimental confirmation of GQ formation, selecting the QGRS with the lowest (G11) and the highest G-score (G20) in its RNA genome. The fluorescence of ThT and its competitive inhibition by PDS indicate that the corresponding synthetic ribooligonucleotides might indeed adopt intramolecular GQ-folds (Figure 2). Notably, the low-scoring G11 might acquire an unusual two-layer GQ structure bearing a zero-nucleotide loop in combination with a long loop. A similar unconventional two-tetrad GQ was previously found in cellular RNA [44], indicating that this type of fold may have broader significance (Appendix A).

ThT fluorescence and its inhibition by PDS suggest that both ribooligonucleotides form all-parallel GQs in K^+^ as well as Na^+^ buffers (Figure 2 and Appendix A). This is the most common conformation of naturally occurring RNA GQs [87]. In contrast to many other examples [56], the thermal stability of G11 and G20 was only moderately enhanced (by ~4–5 °C) by K^+^ as compared to Na^+^. G11 and G20 also differed little in their stability with respect to the same coordinating cation, and they remained intact above 34 °C, the optimal temperature for RV-A2 replication [61]. Of note, similarly to RV-A2, the SARS-CoV-2 genomic RNA contains only predicted 2-layer GQs, and analogous tests showed that they remained stable at a physiological temperature [31,88].

Our analyses with the GQ-binding PDS and PhenDC3 strongly suggest that GQs can form in the context of the encapsidated RV genome. As shown in Figure 3, RV-A2 loaded with PDS was compromised during uncoating. Since PhenDC3, a chemically distinct GQ-interacting compound [89], had the same effect, the targets were in all likelihood G-rich sequences in the genomic RNA that readily formed GQs upon binding of these compounds. A PaSTRy assay showed a markedly enhanced capsid mobility of PDS-loaded virions when compared to the untreated control. This resembles the enhanced capsid dynamics seen upon low-level modification of the encapsidated RNA genome of flock house virus, RV-B14, and Foot-and-Mouth Disease virus [90]. The augmented conformational fluctuation was suggested to originate from disrupted capsid protein-RNA interactions in the native particle [91]. By analogy, we propose that PDS similarly triggers rearrangement of the encapsidated rhinoviral RNA affecting contacts with the inner surface of the shell [7], likely including recently identified enterovirus packaging signals [92].

Our ThT data in Figure 2 might be taken to indicate that the majority of putative QGRS present within the RV genome is in a metastable state with the potential to form the more stable GQs. This is supported by the large-scale ultrastructural reorganization of PDS-treated protein-free RNA cores in the presence of Na^+^ (Figure 4B,C and Appendix A). PDS was previously found to lower the activation energy for GQ formation [93]. The structural change resulting from the postulated PDS-induced refolding must be due to a drastic modification of short- and long-range interactions. This clearly confirms a bioinformatics prediction of the effect of ligand-mediated GQ formation based on transcriptomics-wide QG identification [79]. Importantly, ThT, at the used concentration, does not induce the conversion of hairpins into GQ [94]. An engineered mRNA forming a metastable hairpin could switch into a GQ, using N-methyl mesoporphyrin (NMM) as a light-up indicator [95]. Such switches are increasingly engineered for use in chemical sensors. Our data demonstrate the relevance of this phenomenon for a naturally occurring RNA molecule.

We conclude that during viral positive-strand RNA synthesis in the infected cells, sequences of the nascent (+) strand comprising QGRS probably fold faster into alternative, long-lived metastable conformations (tens of microseconds for a hairpin [96]) than into the more stable GQ (hundreds of milliseconds [97]) as soon as they emerge from the active center of the viral RNA replicase, followed by their rapid encapsidation (which is tightly coupled with replication [98]).

In Na^+^ but not K^+^ phosphate buffers, PDS induced large-scale conformational changes in protein-free viral RNA cores. At first glance, this appears paradoxical as K^+^ usually stabilizes GQ better than Na^+^ (as also observed in our study with the GQ-forming ribooligonucleotides; Figure 2). The underlying cause of this phenomenon is most likely rooted in the tighter compaction of the viral RNA and the strengthening of its tertiary structure in the presence of K^+^, as demonstrated by our ThT assay. This seemingly restricts the access of PDS to its target sequences in the RNA core, either within the protein shell or in the protein-free RNA core. In support of our findings, the degree of RNA compaction (also controlled by the concentration of Mg^2+^) was found to critically regulate the binding of small molecules to the c-di-GMP riboswitch [99]. Water molecules from the first hydration shell interact more strongly with Na^+^ than K^+^ [100], which can influence the compaction and overall RNA 3D arrangement, as extensively discussed in Auffinger et al. [101]. The substantially higher stability of the tertiary fold in the presence of K^+^ compared to Na^+^ furthermore suggests the presence of structure-stabilizing sites specifically chelating the larger K^+^. Structurally important K^+^-specific pockets have been described for ribozymes and ribosomal RNA [102,103]. The unexpected ‘protective’ effect of K^+^ was also clearly evident when RV-A2 was incubated with PDS, and mass spectrometry revealed PDS accumulation only in the presence of Na^+^. Consequently, the latter also resulted in a profound drop in infectivity, as observed for viruses incubated with PDS or PhenDC3 in Na^+^ phosphate buffer. One might consider testing this ‘compactness’ via incubation of viral RNA cores in Na^+^ and K^+^ phosphate buffers, respectively, and assessing their sensitivity toward RNase digestion.

How does the binding of PDS affect RNA release from the capsid? The current model of enterovirus RNA exit favors the transient unfolding of the RNA to pass through one of the small pores [9] in the expanded subviral A-particle. The emerging RNA would then arrive in the cytosol without ever contacting endosomal content [104]. The average number of PDS molecules (~10) incorporated into the capsid of RV-A2 in the presence of Na^+^ is comparable to the number of putative QGRS (11). Assuming that these QGRS exist as folded GQs, it is likely that the stabilizing effect of PDS prevents their unwinding. With an effective diameter of ~3.6 nm for the RNA Q-quadruplex [105], the GQ-ligand complex would constitute a roadblock to RNA transfer similar to that acting on transcription and translation mentioned above in the context of other viruses and even more similar to psoralen-cross-linking of rhinoviral RNA [12]. The structural reorganization of the encapsidated genome triggered by PDS, as indicated by PaSTRY and directly visualized in ex-virion RNA, might also compromise the well-ordered RNA layer beneath the protein shell of the A particle proposed to guide its ordered egress [7]. Alternatively, it might additionally dislodge the viral RNA’s 3′ end found to exit first [11] from a position believed to reside in the vicinity of one of the pores opening at the two-fold axis. PDS-driven relocation would result in a high entropic penalty for finding such holes via thermal fluctuation, critically diminishing the successful vectorial traversal of the viral RNA through the capsid wall. The importance of proper 3′-end positioning was demonstrated in a coarse-grained model of RV-A2 uncoating [106]; our results with PDS now provide initial experimental evidence that supports this mechanism. This does not contradict the fact that heat-triggered uncoating of RV-A2 is unaffected by PDS, as the substantially increased thermal motion will restore the RNA’s chances of finding a suitable exit pore. An alternative model of enterovirus uncoating proposes the cracking of the capsid at low pH, enabling the exit of the viral RNA in bulk without the need for substantial unfolding [16,17]. While PDS might also interfere with such a process, we consider this mode of uncoating less important for RV-A2 as cryo-EM analysis of low pH treated RV-A2 revealed only low amounts of open particles, i.e., those lacking one or more pentamers as end products of uncoating, whereas many more empty particles were closed [7].

Time-of-drug-addition experiments showed that PDS had little consequence for RV-A2 protein production when added after the uncoating stage. All subsequent events required for viral reproduction occur in a high K^+^ and low Na^+^ cytosolic environment, limiting the access of the compound to its target sequences within the viral RNA, as found for the ex-virion RNA. However, a complete block of PDS binding by high intracellular K^+^ is unlikely, based on recent life cell imaging of cellular mRNA GQs in the absence or presence of a variant of PDS [107]. Previous experiments with four-layer GQs demonstrated their unwinding by the DHX36 helicase, also when stabilized by GQ ligands such as PDS and PhenDC3. We thus believe that the battery of intracellular host cell-derived helicases [108], together with the virus-encoded helicase 2C [109] efficiently disrupt intracellularly formed, intrinsically weak two-layer rhinoviral GQs even when bound by PDS. The need for facile replication may explain why the vast majority of RVs lack putative QGRS predicted to form three-layer G-quadruplexes that might disrupt this process.

In summary, we have shown that targeting putative QGRS by GQ-stabilizing compounds specifically inhibits the uncoating of a common cold virus. We provide a mechanistic explanation for this result using biophysical and ultrastructural analyses. Strikingly, PDS uptake into the virus occurs in the presence of Na^+^ but not K^+^ due to their differential impact on viral RNA compaction rather than GQ formation. While PDS has potential side effects that may preclude its direct use as a therapeutic, our data point to GQ stabilization as a novel therapeutic avenue for tackling rhinovirus infection. Finally, PDS did not interfere with RV-A2 binding to the host cell and likely preserved the immunogenic epitopes, making these weakly infectious particles attractive candidates in the development of attenuated vaccines.

## Figures and Tables

**Figure 1 viruses-15-01003-f001:**
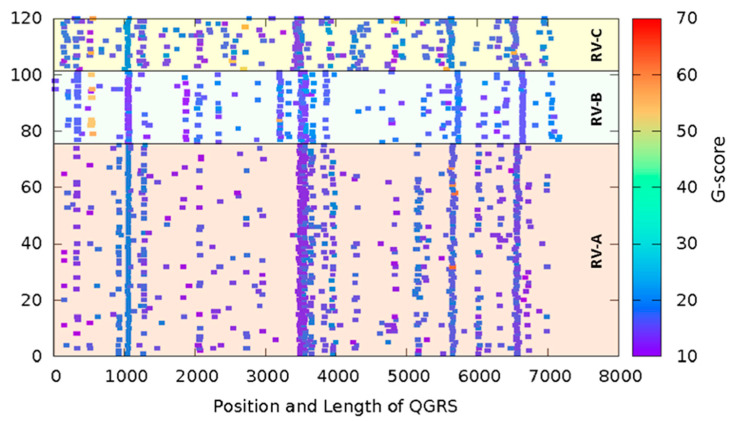
All RV genomes harbor conserved sequences with propensity to form GQs. Complete RNA sequences of 120 RVs available in the NCBI database were analyzed for putative GQ-forming sequences using the QGRS mapper software. G-scores ≥ 10 are shown as rectangles with their lengths corresponding to the number of bases making up the putative GQ. The color bar indicates the respective G-scores. Note that the sequences were used without alignment optimization, i.e., neither deletions nor insertions were taken into account. Also, note the four vertical lines indicating strong conservation. Species RV-A, RV-B, and RV-C are indicated.

**Figure 2 viruses-15-01003-f002:**
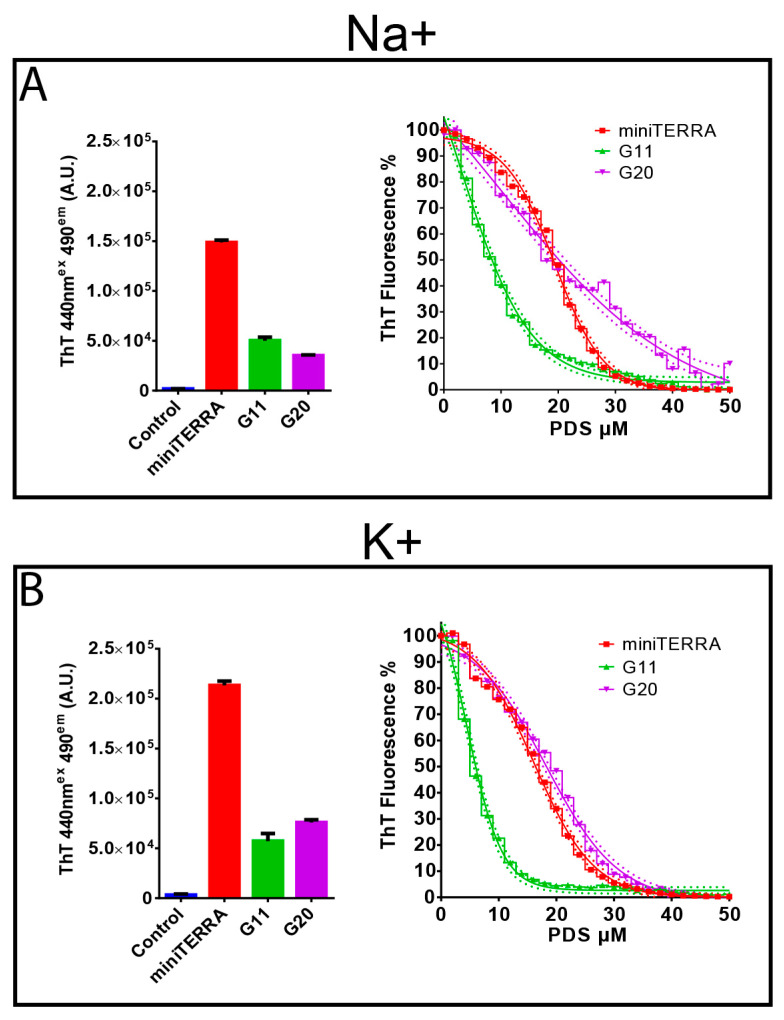
Impact of Na^+^ and K^+^ on ThT binding and ThT displacement by PDS. Ribooligonucleotides dissolved at 5 µM in 100 mM Na^+^ (**A**) or K^+^ (**B**) phosphate buffer (both at pH 7.4) as indicated, were incubated with ThT, and the fluorescence was measured at 490 nm (*n* = 3; left panel). Right panels; as above, but fluorescence was measured in the presence of increasing concentrations of PDS added in steps. Values are normalized to the initial fluorescence signal (i.e., without PDS = 100%; *n* = 3).

**Figure 3 viruses-15-01003-f003:**
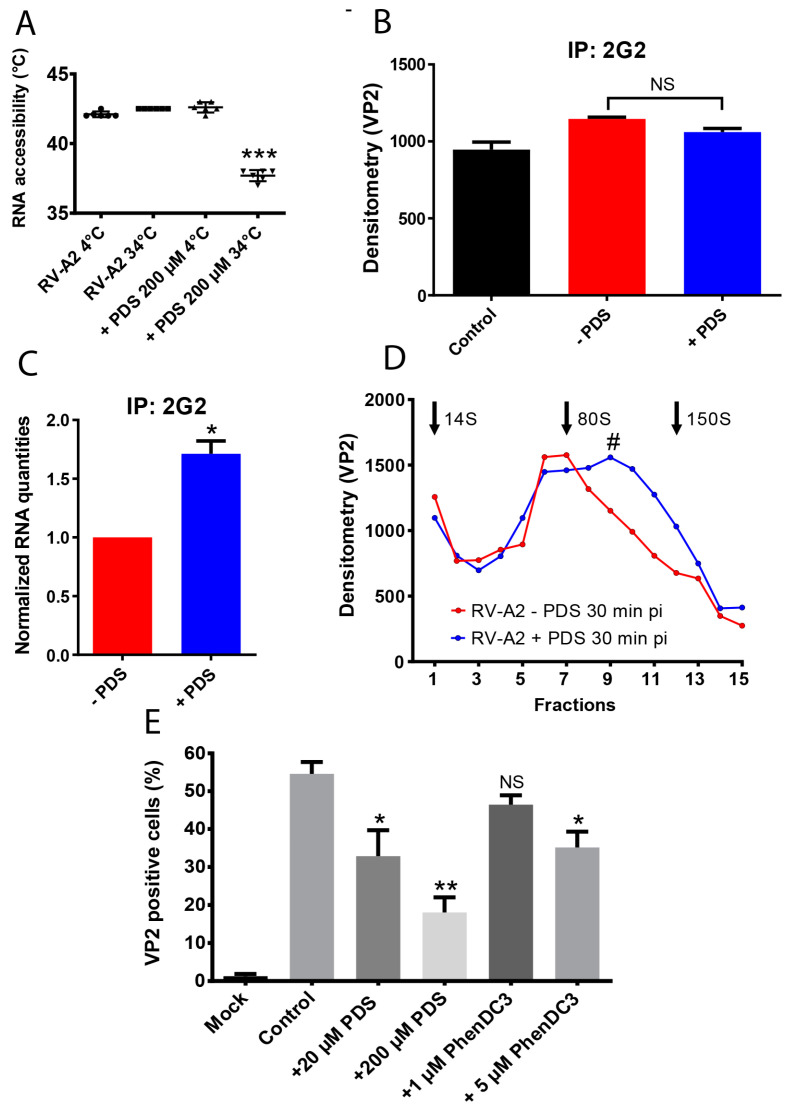
GQ-binding compounds decrease the temperature of RV-A2 capsid permeability onset, impair uncoating, and decrease infectivity. (**A**) Scatter plot of the temperature of onset (T_on_) of capsid permeability (i.e., RNA accessibility for SYTO 82) determined by PaSTRy with purified RV-A2 that had been pre-incubated +/−200 µM PDS at 34 °C for 4 h. ‘***’, note the strong reduction in the temperature of RNA accessibility-onset upon pretreatment with 200 µM PDS at 34 °C. (**B**,**C**) HeLa cells were infected with purified RV-A2 pre-incubated +/−PDS as in (**A**). Subviral A- and B-particles were immunoprecipitated with mAb-2G2. (**B**) Viral protein was quantified by using mAb 8F5 and a goat anti-mouse IgG HRP-conjugated secondary antibody, followed by measuring VP2 by densitometry. (**C**) RV-A2 RNA levels were quantified by qPCR and normalized relative to a known amount of AiV seed virus spiked into samples prior to RNA isolation. (**D**) Cleared supernatants prepared as above were separated by 10–40% (*w*/*v*) sucrose density gradient ultra-centrifugation. Dot blots were prepared from each fraction, and VP2-containing viral material was quantified with mAb 8F5 and IRDye 680RD goat anti-mouse IgG secondary antibodies (*n* = 3). The obtained signal intensity was plotted against fractions from top to bottom. Note that VP2 is present in all viral and subviral particles. Native virus (150S) and subviral B-particles (80S) were used as sedimentation controls and run on separate gradients. Their positions are indicated in the plot; the position of 14S pentamers was inferred from the literature. ‘#’ indicates the approximate position of 135S A particles (**E**) Purified RV-A2 was incubated with +/−PDS or PhenDC3 at the concentrations indicated for 4 h at 34 °C. Free compounds were removed by centrifugal ultrafiltration, and the retained material was used to infect HeLa cells. The percentage of infected cells was determined at 9 h pi by FACS analysis of the intracellularly produced VP2 with mAb 8F5 and an anti-mouse AlexaFluor 488 conjugated secondary antibody and is indicated as bars (*n* = 3; * *p* ≤ 0.05; ** *p* ≤ 0.01).

**Figure 4 viruses-15-01003-f004:**
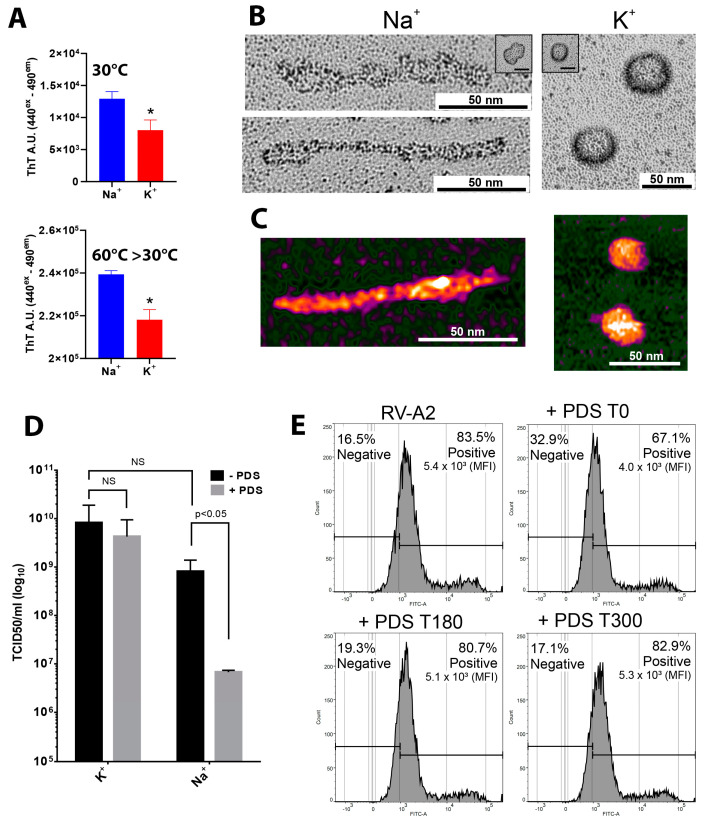
PDS treatment of protein-free RNA cores results in a drastic reorganization in the Na^+^ but not the K^+^ phosphate buffer. (**A**) The RV-A2 capsid proteins were digested via gentle proteolysis in 100 mM Na^+^ phosphate buffer or 100 mM K^+^ phosphate buffer, as indicated. The protein-free RNA cores were then mixed with ThT (final concentration of 5 µM), and the ThT fluorescence signal was acquired at 30 °C. (**B**) RV-A2 RNA (prepared as in (**A**)) was incubated with 20 µM PDS for 10 min at room temperature and subjected to rotary shadowing followed by TEM. Insets depicting representative images of non-PDS-treated ex-virion RNA in Na^+^ or K^+^ phosphate buffers are displayed. (**C**) Samples similarly treated as in (**B**) were analyzed by AFM in the presence of 100 mM Na^+^ or K^+^ phosphate buffer. (**D**) Purified RV-A2 was diluted in the above buffers and incubated overnight with or without 20 µM PDS at room temperature. After ultrafiltration, the infectivity of the respective retentates was determined by end-point titration. Viral titer is expressed as TCID_50_ in the bar graph (*n* = 3). The significance of the differences was evaluated by the two-way ANOVA; NS—statistically not significant (*p* ≥ 0.05). (**E**) HeLa cells were challenged with RV-A2 (MOI = 10) for 30 min at 4 °C, allowing virus attachment. A synchronized virus entry was triggered by a transfer to 34 °C. Immediately before (=0 min pi), 180 min, and 300 min pi after the temperature shift, the medium in the respective well was adjusted to 20 µM PDS, and incubation of cells continued for 9 h to allow for one cycle of infection. Non-infected cells without PDS treatment were used as controls. Cells were immunostained with mAb 8F5 and analyzed by flow cytometry. Non-infected and infected populations are displayed at the left and right, respectively, in the fluorescence intensity histogram, and the corresponding percentage is provided on top. MFI is the mean value of fluorescence intensity calculated for each sample. ‘*’ *p* ≤ 0.05.

**Table 1 viruses-15-01003-t001:** Sequences of synthetic ribooligonucleotides, their positions on the RV-A2 genome, and G-scores are in parentheses. Note that in the negative control, all Gs of miniTERRA are replaced by Cs.

Ribo-Oligonucleotide/(G-Score)	Sequence
Negative control/(0)	5′-UUA CCC UUA CCC UUA CCC UUA CCC UUA-3′
miniTERRA (42)	5′-UUA GGG UUA GGG UUA GGG UUA GGG UUA-3′
G11; base # 2048–2074 (11)	5′-GGC ACU CAU GUU AUA UGG GAU GUG GGG-3′
G20; base # 1038–1064 (20)	5′-CCU CAA AGG GUU GGU GGU GGA AAC UAC-3′

**Table 2 viruses-15-01003-t002:** IC_50_ values for the displacement of ThT by PDS. These assays suggest an intrinsic ability of the two RV-A2-derived QGRS to form two-layer RNA GQs.

Ribo-Oligonucleotides	IC_50_ (µM)	
	Na^+^	K^+^
mini TERRA	19.29 ± 0.25	17.23 ± 0.28
G11 (position 2048–2074)	5.20 ± 0.18	9.11 ± 0.29
G20 (position 1038–1064)	17.13 ± 0.45	19.38 ± 0.32

**Table 3 viruses-15-01003-t003:** Mass spectrometry quantification of PDS (moles per mole virus).

Treatment	PDS Bound (Moles/Mole Virus)	Relative to Maximum
34 °C/Na^+^	10.0	100%
4 °C/Na^+^	0.9	8.8%
34 °C/K^+^	0.2	1.4%
4 °C/K^+^	0.4	3.7%

## Data Availability

Data can be obtained from the corresponding authors.

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
