# Peer review of "Stabilization of the Quadruplex-Forming G-Rich Sequences in the Rhinovirus Genome Inhibits Uncoating—Role of Na+ and K+"

_viruses, 2023, doi:10.3390/v15041003_

Round 1
Reviewer 1 Report
Comments to Real-Hohn and colleagues ‘Stabilization of Quadruplex-Forming G-rich Sequences in the Rhinovirus Genome Inhibits Uncoating – Role of Na+ and K+’. This is an interesting paper, with relevance for virology, and beyond. The authors provide strong evidence that rhinovirus A2 genomic RNA contains 4-stranded higher-order intramolecular guanine-rich stretches, so called G-quadruplex (GQ) secondary structures. These GQs can be targeted by small chemical compounds, particularly in presence of sodium ions. This creates a situation that renders the virions uncoating defective.
Specific comment:
I congratulate the authors to this comprehensive study. The paper is well written, and results are carefully documented and controlled. The paper is ready for publication, if a minor point can be addressed by experiment or by a written clarification.
In figure4A, can the authors perhaps provide some kind of evidence to show / discuss the efficacy of protease treatment under Na+ and K+ conditions, respectively, e.g., by spiking in a fluorogenic protease substrate, or providing arguments to support the notion that proteolysis is similarly effective under Na+ and K+ conditions? Reason for bringing this up is that the differences between the Na+ and K+ conditions seem to be rather subtle in terms of ThT fluorescence.
Author Response
We thank this referee for his/her constructive suggestions!
Incubation of the virus with a proteinase in Na+ vs K+ containing buffers to assess the 'openness'/flexibility of the capsid is an excellent idea. Indeed, comparing the sensitivity of native virus with that of A- and B-particles has been done and shown a much increased sensitivity of the latter two toward proteolysis. However, a possible influence of the type of cation present has not been assessed. With respect to the RNA genome, we do not expect RNase to enter the virion or subviral particles precluding a similar experiment with RNase. However, we might assess the degrees of 'openness' or compactness of the RNA core. So, the logic experiment would rather involve RNase digestion of viral RNA cores in Na+ and K+ phosphate buffer +/- PDS followed by determination of the degree of RNA cleavage. Unfortunately, due to my retirement we cannot carry out such experiments any more. However, we added a sentence to the text mentioning the possibility of assessing the influence of Na+ v/s K+ buffers on RNA digestion of protein-free cores.
We again checked the spelling and made a number of corrections to the grammar and style and corrected typing errors. As also pointed out in our reply to referee 2 we noticed a misnumbering of the references starting with #57. We also renamed some of the chapters to minimize confusion. By the same token we deleted Fig. S5 plus corresponding text since it did not very much contribute to the overall conclusion but might have led to misunderstandings. We have corrected all the above to the best of our knowledge and believe that the accordingly revised manuscript is now ready for publication.
Reviewer 2 Report
Stabilization of Quadruplex-Forming G-rich Sequences in the Rhinovirus Genome Inhibits Uncoating – Role of Na+ and K+.
Antonio Real-Hohn et al.
This is an interesting study by respected authors, the main claims of the paper are generally supported by the data and it would therefore be good to see this published. However, some sections are poorly written, very technical and difficult to understand for the reader of a virology journal. I am an experienced virologist and for several sections I was not able to understand sufficiently to review the data and in my opinion the writing in these sections should be improved.
Abstract
The abstract does not provide an easy to understand summary of the work.
I find these two sentences especially difficult to understand and recommend they are improved.
1. “Focusing on the genomic RNA as a possible target for antivirals we asked whether impacting on sequences predicted to form G-quadruplexes (GQs) might inhibit viral propagation.”
Is ‘impacting’ the correct word?
What are G-quadruplexes (GQs)? I think these should be described in non-expert terms in the abstract.
2. “Synthetic RNA oligonucleotides with sequences corresponding to the highest and lowest GQ score indeed formed GQs.”
What is a GQ score?
It is not clear how the properties of a synthetic RNA are related to what is happening in the virus?
Main text
Introduction
Line 63: suggest using ‘safe’ (not ‘save’)
Line 80-84 (description of GQs): I highly recommend adding a non-technical explanation of GQs. Would a figure help to explain this? There is a figure in the supplementary data which I only saw later, is this intended to provide this explanation and if so should it be in the main paper? What is Hoogsteen hydrogen bonding? What are π-π interactions? I understand what RNA secondary structure is but after this introduction to GQ I still don’t have a clear idea of what they are…
Line 101-103 (GQ-stabilizing compounds inhibit viral replication): please make clear which viruses and references this statement applies to.
Results
3.1 This sounds like a technical description of the ‘rationale’ for how the analysis was done. But I did not understand what has actually been done, what results were generated and how they were analyzed. A figure is shown and there is mention of conservation of signal in the figure legend, hinting at some significance to this pattern (after I studied the figure and figure legend I can understand there are interesting patterns in the data) but I don’t understand what they mean and there is no ‘description’ of the results in the main text! Please describe in the text. What is the G score and what is the significance of the G score? Are the orange/yellow boxes in RVB and C sequences important? Why do some RVA sequences have extremely high G score boxes around position 5500 when most viruses have much lower G scores? These differences could undermine the confidence in this analysis if they are not explained. Perhaps it is covered later in the discussion but more explanation is required in the results section in order for the reader to understand the results.
3.2 Section title: specific binding between what and what??? What are RNA sequences G11 and G20? What is the principle of this assay? What is miniTERRA? I don’t understand any of this section or figure 2. Please improve how this is explained.
3.3 Introduction to this section is clear. But 401-407 text confuses between breathing, ability of drug to penetrate capsid and the effect on virus stability and uncoating. The data in Fig 3A looks very clear when you understand the figure but the actual results are not described in the main text, please add a description. In contrast, the supplementary data is described - if the supplementary data is essential to make the data understandable then why is it not shown as a figure in the main paper?
The text corresponding to Fig 3B and 3C and 3D seems much clearer.
Fig 3E and corresponding text: ok but I have some concerns about using flow: What is the time for a single replication cycle of RV? What time post infection does CPE begin? What state will infected cells be in at 9 h pi? I would expect cells infected with a picornavirus (enterovirus) to be dead or almost dead at 9 hrs post infection, so how can they be analyzed by flow? The method starts with approximately 1,000,000 cells and ends with only 104 events so the entire process is only analyzing a tiny proportion of the sample with a tiny number of events. This reinforces my concern about loss of sample due to fragile condition of infected cells. Why was a conventional assay for infectivity (PFU or TCID) not used (as in the next figure)? Please add details to remove these concerns and show supplementary data showing numbers of cells at each step, flow gating strategy etc.
3.4 and Fig 4. Another confusing section. line 484 what are protein-free virion cores? How were they generated? ‘increased fluorescence emission’ in what assay? Is this the same assay I did not understand in section 3.2? If I have understood correctly then the finding that signal for GQ increased after heating ex-virion RNA to 60C is interesting. What is the principle of rotary shadowing? What are platinum replicas? I have not read the methods yet! Presumably this is covered in the methods but is it possible to make the results section understandable without constant reference to the methods? Why is the DSF only shown in supplementary data?
When figure 4 has been studied and eventually understood it mostly looks ok except two concerns:
i) it is not clear if the data in 4D is the direct titration of virus infectivity in the samples (this would be the expected approach) or if the samples have been re-amplified by growth in culture before titration (this was alluded to in lines 593-595 saying ‘challenge of cells’ and ‘final viral yield’) I don’t understand this second approach.
ii) as with the previous concerns about flow cytometry I would need more convincing that flow as used for 4E is the best assay for this experiment. Why is simple titration of infectivity not used?
Discussion
There are some good discussion points. Parts of the discussion also help to explain some of the difficult to understand results sections but I think the results section should be understandable in isolation. I still don’t understand what the work with the synthetic RNAs is meant to show. Parts of the discussion are also very technical e.g. I still don’t understand enough about GQs to appreciate the significance of 2 layer versus 3 layer GQs…
Author Response
We thank this referee for his/her detailed analysis of our text and the recommendations for improvement. Here is a point-to-point list of our answers and the modifications we made to the text:
1) Due to the journal-imposed limitations of the number of words in the Abstract we did not dwell much on G-quadruplexes here but rather amply explained and referenced them in the main text. See e.g. in the Introduction: "Among these secondary structures are G-quadruplexes (GQs), 4-stranded higher-order folds formed intra- or inter-molecularly by guanine-rich stretches of DNA or RNA. The basic unit consists of a square, co-planar arrangement of four guanines (a tetrad or G-quartet) connected via Hoogsteen hydrogen bonding. GQs with at least two layers result from self-stacking via π-π interactions"..... However, following this reviewer's request we now explain, in the Abstract, the meaning of G-quadruplexes (QGs) with some additional 80 words. Furthermore, we changed 'impact' to 'inhibit', which is better explaining one of the objectives of our study.
2) As mentioned, the GQ score is now shortly explained in the Abstract. If a synthetic oligonucleotide forms a GQ it is very likely that a GQ can also form from the same sequence when it is part of the genomic RNA. So, the synthetic ribooligonucleotides were derived from the RV-A2 sequence and tested for their propensity to form GQs. We believe that the above modifications of the Abstract makes this now clearer.
Introduction:
As indicated in the response to the other referees, we have carefully revised the text for typing errors and also corrected save > safe. Hoogsteen base paring is a non-canonical paring of guanines. This, and π-π interactions are well-explained in Molecular Biology text books as well as in Wikipedia. Of note, there is also active research on the use of them in sensors for metal ions and small molecules. We thus do not believe that it is necessary to explain it here.
We have changed 'GQ-stabilizing compounds inhibit viral replication' to 'GQ-stabilizing compounds inhibit replication of various viruses'. Papers on viral inhibition by such compounds are referenced several times in the text. For example see: ".... GQs are also found in the genome of DNA and RNA viruses such as Ebola virus, herpes simplex virus, human papillomavirus, human immunodeficiency virus, Zika virus, Influenza virus, human coronaviruses, and hepatitis C virus [29-36], where they control various steps in the virus life cycle, ranging from protein expression and nucleic acid replication to assembly into nucleocapsids [37]..... Going into more detail would have further increased the length of this already quite long paper.
Results: As mentioned above, the GQ score is explained now in the Abstract and again in the text and also in various papers referenced in our manuscript. Since research on G-quadruplexes is very active (Medline yielded 8051 hits upon searching for 'G-quadruplex' and 291 hits upon searching for 'G-quadruplex & virus' on the 11th of April) we believe that the scientific community is sufficiently aware of these terms. Please also note that, in the Methods section, the software and the procedure for determining the GQ-score is described in detail.
There is a number of other points raised by this referee. Most refer to the significance of GQ and clarity of the text. He/she drew our attention to some misleading errors in the text and some passages resulting in confusion. We have adjusted entire passages accordingly and also deleted Fig. S5 as it did not contribute much to the general conclusion. We believe that all these modifications now satisfy this referee's requests and the revised manuscript is now ready for publication.
Reviewer 3 Report
In this manuscript, Real-Hohn and co-authors characterize RNA G-Quadraplexes in rhinoviruses, They provide studies on 2 synthetic ribo oligonucleotides as well as on encapsidated or capsid-liberated RV-A2 genomic RNA. Strengths of this manuscript include clear writing, highly detailed methods text, and that they typically used more than one different technique/assay when testing the same component of their hypotheses. A further strength is conservative wording in how they interpreted their results. I could find nothing factually inaccurate. My only suggestion for improvement is a minor one: in the results section, the reader is sometimes primed with speculative statements about how to interpret the data. Some of this (e.g. lines 587-592, 607) may be better placed in the Discussion section or simply trimmed from the manuscript, or at a minimum, perhaps check the use of “might” in the Results section.
Author Response
We thank the referee for his/her kind words and constructive suggestions!
We are aware of the speculative nature of some of our statements. Following this referee's suggestion we thus changed too strong claims for weaker statements by using the words 'might' and 'could' and so on. However, we decided to not move any of the explications and interpretations from the Results section to the Discussion section as we believe that some redundancy aids understanding and keeping track as they lead over to the next paragraph.
With respect to the first referee we indeed found a number of typing errors, in particular lacking superscripts that seriously compromise the meaning. During this re-reading process we also noticed a misnumbering for the references starting from #57. All this has been corrected to the best of our knowledge. The more extensive changes are explained in the response to referee #2.
We believe that the revised manuscript in its present form is now ready for publishing.